# Material-Inherent Noise Sources in Quantum Information Architecture

**DOI:** 10.3390/ma16072561

**Published:** 2023-03-23

**Authors:** HeeBong Yang, Na Young Kim

**Affiliations:** 1Institute of Quantum Computing, University of Waterloo, 200 University Ave. West, Waterloo, ON N2L 3G1, Canada; 2Department of Electrical and Computer Engineering, Waterloo Institute for Nanotechnology, University of Waterloo, 200 University Ave. West, Waterloo, ON N2L 3G1, Canada; 3Waterloo Institute for Nanotechnology, University of Waterloo, 200 University Ave. West, Waterloo, ON N2L 3G1, Canada; 4Department of Physics and Astronomy, University of Waterloo, 200 University Ave. West, Waterloo, ON N2L 3G1, Canada; 5Department of Chemistry, University of Waterloo, 200 University Ave. West, Waterloo, ON N2L 3G1, Canada

**Keywords:** quantum information processing, quantum technologies, noise, decoherence, superconducting quantum system, semiconductor spin-qubit quantum systems, trapped-ion quantum systems

## Abstract

NISQ is a representative keyword at present as an acronym for “noisy intermediate-scale quantum”, which identifies the current era of quantum information processing (QIP) technologies. QIP science and technologies aim to accomplish unprecedented performance in computation, communications, simulations, and sensing by exploiting the infinite capacity of parallelism, coherence, and entanglement as governing quantum mechanical principles. For the last several decades, quantum computing has reached to the technology readiness level 5, where components are integrated to build mid-sized commercial products. While this is a celebrated and triumphant achievement, we are still a great distance away from quantum-superior, fault-tolerant architecture. To reach this goal, we need to harness technologies that recognize undesirable factors to lower fidelity and induce errors from various sources of noise with controllable correction capabilities. This review surveys noisy processes arising from materials upon which several quantum architectures have been constructed, and it summarizes leading research activities in searching for origins of noise and noise reduction methods to build advanced, large-scale quantum technologies in the near future.

## 1. Introduction

With inundated information, we encounter an urge to discern accurate facts, essential knowledge, and necessary evidence in an effective way. Keeping up with the growing reliance on the information in our modern society, the demand for advanced, competent technology to compute and communicate information is higher than ever, and meeting this demand involves a series of activities to engender, manipulate, store, transfer, and receive information amid numerous parties. High-speed and high-fidelity computation and communication become inevitable in the retrieval of necessary and useful information from the overwhelmingly massive amount of data around us, and unprecedentedly versatile technologies will be pivotal to supporting innovation-based winning societies in the twenty-first century. Hence, the realization of high-performance computing (HPC) has been a sought-after goal in academia, industries, and government laboratories. Among the many HPC architectures, quantum technologies possess the promising potential to offer radical advancement in computation, communications, simulations, and sensing by exploiting governing principles of quantum *entanglement, superposition, coherence*, and *tunneling* [1]. Indeed, quantum information processing (QIP) starts from the definition of a qubit, a series of information encoding and decoding steps on qubits via multitudes of operations, and the dissemination of information between parties quickly, correctly, and safely.

Information in the classical world is expressed as sequences of binary states, e.g., on/off or bits of 0/1, whereas a *qubit* is a basic unit of quantum information and represented by a two-level quantum system, whose base states are denoted as |0〉 and |1〉. A famous example of solid-state materials is spin degrees of freedom, which can be either up or down. Although its binary nature resembles classical bits, the qubit is ultimately different in that its state can have a potentially infinite number of possibilities with |0〉 and |1〉 before measurement. Mathematically, a quantum state |ψ〉 is expressed as the linear superposition of two base states,
|ψ〉=α|0〉+β|1〉,
where arbitrary complex values of α and β have both amplitudes and phases. In a pictorial way, the qubit state is also visualized on a so-called Bloch sphere, whose north and south poles are at |0〉 and |1〉, respectively, and the arbitrary state |ψ〉 on the surface is a vector determined by two angles, θ and ϕ, as illustrated in Figure 1a. Without loss of generality, we express it on the basis of the *z*-axis, which is called longitudinal, and the *x* and *y* axes are transverse.

The measurement of |ψ〉 is used to find the probability of being at |0〉 and |1〉 from the amplitude squares of α and β, respectively. As far as the phase of |ψ〉 is concerned, the relative phase between α and β is observable, especially when the qubit state evolves over time. The ability to reveal remarkable and subtle quantum natures such as quantum *coherence* is unique, and the change of phase is portrayed as the rotation of |ψ〉 on the Bloch sphere. As a matter of fact, QIP is engaged in controlling the state of individual qubits or two interacting qubits, which are known as quantum gates. In an ideal situation, when qubits only respond to designed gate operations, they end up being in the intended states, yielding the perfect fidelity of QIP. Unfortunately, in reality, qubits are placed in an environment and their amplitudes and phases are modified externally by known or unknown reasons via the interactions of the environment. Hence, the qubit states become fluctuating or *noisy*. Figure 1b–d sketch two possible routes to lose original information: one is the energy relaxation process quantified by $T_{1}$, a characteristic longitudinal relaxation time from a higher energy state to the ground state or vice versa given by the inverse of $\Gamma_{1}$ (Figure 1b), or its phase can be lost with $T_{\phi }=1/\Gamma_{\phi }$ due to longitudinal noise sketched in Figure 1c. After a certain time $T_{2}$, which is known as the transverse relaxation time or decoherence time, it concerns both longitudinal relaxation time and pure dephasing time $1/T_{2}=\Gamma_{2}=\Gamma_{1}/2+\Gamma_{\phi }$. A successful gate operation requires a long $T_{1}$ and $T_{2}$ so that any unwanted energy or phase relaxations of qubits due to various noise sources are small or negligible. Since the gate fidelity is indeed inversely proportional to coherence time, longer $T_{1}$ and $T_{2}$ will facilitate the high fidelity of gate operations essential for fault-tolerant quantum computation.

*Entanglement* among quantum states is idiosyncratic and has no classical counterpart. When two particles are entangled, the information of one particle is fully sufficient to know the information of the other particle, which is the ultimate correlation between the two. Mathematically, the entangled states are not a product state of the constituents; in other words, the entangled states are not separable. One of the classic entangled states is the Bell state of two qubits:$$|\Phi^{+}\rangle =\frac{|00\rangle +|11\rangle }{\sqrt{2}},$$
$$|\Phi^{-}\rangle =\frac{|00\rangle -|11\rangle }{\sqrt{2}},$$
$$|\Psi^{+}\rangle =\frac{|01\rangle +|10\rangle }{\sqrt{2}},$$
$$|\Psi^{-}\rangle =\frac{|01\rangle -|10\rangle }{\sqrt{2}},$$
where none of the Bell states are expressed as the product states of each qubit state. Entangled particles were indeed advantageous to demonstrate quantum cryptography and quantum teleportation in communications [3,4,5].

Quantum information science and technologies are anticipated to be transformative in computation, communications, simulation, and sensing. Four decades ago, Richard P. Feynman envisioned that processors governed by quantum mechanical rules would outperform classical computers in solving complex problems [6]. Examples of complex problems include quantum many-body physics [7], the complex theory of disordered spin-glass phase [8,9], mysterious brain cognitive function [10,11], folding and unfolding DNA in biological systems [12], and molecular dynamics in quantum chemistry [13]. For the last several decades, prototypical quantum technologies have been demonstrated in several notable material systems, primarily led by academic groups under governmental and industrial support. Now is an interesting time, as lab-level quantum technologies in universities are steadily expanding commercial quantum products toward a large-scale deployment of quantum computation and quantum communications led by global industries and start-up companies. Commercial quantum computers and cloud services are now available and are used for many applications in fundamental studies of physics, complex traffic controls, financial sectors, medical divisions, and energy challenges [14]. Despite such impressive and remarkable progress, current quantum technologies encounter serious noise issues to shorten coherence time and scaling issues to limit a finite number of qubit arrays. Dr. John Preskill coined the term *noisy intermediate-scale quantum* (NISQ) era to pinpoint the current status of quantum technologies [15]. A critical pre-requisite for large-scale practical quantum architectures is to identify the origins of noise processes and to harness skills and techniques for maintaining the high fidelity of quantum systems by controlling unavoidable noise processes.

History has demonstrated that revolutionary technologies in a certain epoch are enabled by superior materials. Indeed, noticeable progress in both the hardware and software of quantum technologies is also tightly associated with the choice of materials. There are certainly leading material systems to attain such advancement, namely, aluminum-based superconducting qubit systems [16,17] and trapped-ions [18,19]. As new contenders, photon–qubit systems in solid-state materials are becoming commercialized as large-scale quantum hardware [20,21,22]. Other active material systems for quantum technologies are based on semiconductor quantum dots [23,24], but there are many defects in solid-state materials [25,26,27]. Recently, semiconductor quantum emitters have been integrated into nanoplasmonics [28], where emitted light can reach a sub-diffraction limit due to its ultimate localization on a scale of tens of nanometers. A report demonstrating near-field strong coupling in a single colloidal QD in a plasmonic nanocavity at ambient temperatures shows the promise of such a system to be a realistic quantum technology in the near future [29]. When two QDs are embedded in a moveable plasmonic waveguide, theory predicts that single photon emissions occur at ultrafast speed and photon entanglement can be achieved [30].

Regardless of platform, there is a common type of noise that is prevalent and associated with the material itself or interfaces between materials in most solid-state quantum hardware platforms [31,32,33]. To overcome material-inherent noise to realize large-scale, fault-tolerant quantum hardware, there is still a lot of work to be done to improve the multifaceted aspects of materials through design, purification, and fabrication methods, despite the maturity of dominant material systems of quantum technologiess have reached through many decades of research and development. In parallel, researchers are putting tremendous efforts into foraging auspicious material systems in solids and nano-sized structures as a solution to our current limitations on future quantum platforms.

This review focuses on noise properties arising from materials in quantum information architecture, which especially exploits solid-state materials. First, Section 2 introduces the conceptual overview of general noise processes and discusses low-frequency noise, together with its specific type and random telegraph signal. A three-step analysis protocol to investigate multi-level random-telegraph signals is introduced. Then, we examine two solid-state quantum systems of superconducting qubits (Section 3) and semiconductor spin-qubits (Section 4) from the perspective of material-related noise behavior. Recently, surface-trapped ions, a hybrid solid-state and ion system for scalable and compact ion-based quantum architecture, have been developed. Section 5 summarizes research efforts to mitigate noise processes at material interfaces as a limiting factor of trapped-ion decoherence. As the dimension of the basic quantum unit is in the nanoscale or atomic scale, instantaneous switching events among discrete states in a system form multi-level random telegraph signals (RTSs), which have a dramatic impact on device performance and occur frequently in small-scaled devices. Hence, disentangling individual RTSs from complex fluctuations is the first step, followed by the extraction of relevant parameters of the RTSs, which is a challenging task. The final remarks, along with a summary and outlook, are given in Section 6.

## 2. Noise Processes

### 2.1. Fundamentals of Noise

The daily experience of our universe is macroscopic, where we can identify an immense number of objects and systems with our bare eyes. There are a plethora of systems in many fields of biology, chemistry, physics, and engineering. Once a system is specified as an organization of many constituent particles and elements under a special purpose, the environment becomes defined as the rest of the universe, outside of the system. The statistical nature of numerous particles inside a system originates from their own interactions and their coupling to the environment electrically, magnetically, capacitively, or inductively. Scientists and engineers investigate the dynamics of their system of interest by harnessing necessary and novel technologies. In order to study how a macroscopic system behaves over time, repeated measurements are executed in different experimental settings because the signals are often covered by noise, which refers to the spurious temporal fluctuations occurring on top of desired signals. These fluctuations come from various internal and external origins, including equipment sensitivity, imperfect peripheral items of cables and wires, interactions between associated carriers, and interactions between carriers and their environment. While it is easy to identify the causes of noise processes accurately to control noise, a common standard method to examine the system is to rely on statistical analysis such as time-averaging and ensemble-averaging to enhance the signal-to-noise ratio. In addition, the time information in noisy signals is converted to its conjugate frequency domain, where distinctive spectral responses are seen depending on the kinds of noise.

Primarily, the term noise often possesses a negative connotation, referring to the spurious, unwanted, and uncontrollable contamination of a signal. However, there are some occasions where noise processes can be leveraged to achieve positive results. Stochastic resonance in bistable ring lasers or semiconductor devices is a classic example in which a weak coherent input signal is enhanced by adding inherent noise to the input [34]. In cryptography, security and privacy are strengthened by noise, which can scramble small external disturbances such that the output cannot distinguish a small change, which is described as differential privacy [35,36]. The impact of exploiting noise constructively is interesting to be explored; however, it is out of the scope of this review.

Two representative noisy signals in the time domain and their resulting signal distributions are plotted in Figure 2a,b, and the statistical properties of the system under test are characterized by mean, standard deviation, skewness, and kurtosis. Although both noisy examples in Figure 2b obey the Gaussian statistics as a common model of random stochastic processes, non-Gaussian noise processes are also present depending on the origin and the coupling strength of the system with other constituents or the surrounding external world. There are various types of noise for a given system. For instance, some external noise sources can be eliminated by routine and advanced noise canceling or reduction techniques of filtering, grounding, shielding, modulating, phase locking, and more. Even when the system is at equilibrium, there remains non-reducible intrinsic noise which reveals the statistical features arising from enormous numbers of particles inside the system and which sets the fundamental sensitivity limit. In addition, a new type of noise appears when the system is out of equilibrium.

Consider a two-terminal diode in a laboratory as a simple but explicit example. At a non-zero temperature, when the diode is biased at a certain voltage (*V*), it is well-known that the current (*I*) through the diode with resistance (*R*) is fluctuating in time. Although all possible noise reduction techniques are applied to fight against these fluctuations, residual time-variations in the signals still persist, which originate from intrinsic statistic processes [37]. At finite temperatures, thermally agitated electrons collide with lattice vibrations and impurities of the hardware materials, altering the velocity and position of each electron randomly. As a consequence, a fluctuating current appears across the diode, which is called the Johnson−−Nyquist
thermal
noise. This thermal noise can stay even at zero bias, and it is gone only at zero temperature. While the majority of the electrons and holes participate in electrical transport, the current results from emissions of discrete electrons or holes governed by the Poisson statistics, which means that emitted electrons are independent of each other in a Markov process. This is called shot
noise. There is a low-frequency (or 1/*f*) noise, which broadly refers to a phenomenon in which signal fluctuations tend to be greater at lower frequencies. Clear knowledge of its microscopic origin is still lacking at the moment, but it occurs ubiquitously. Depending on the response trend in the frequency, there are other names for pink noise or random telegraph noise. Johnson–Nyquist thermal noise is a typical example of Gaussian noise, whereas a random telegraph noise that follows a geometric distribution is an example of non-Gaussian noise.

Thorough knowledge of the noise types and their origins is closely tied to the limiting performance of devices and systems; consequently, this may provide clues and means to exclude or dampen unwanted noise and to enhance the device’s sensitivity and capacity. Hence, many scientists and engineers have established systematic mathematical frameworks and various models to handle dynamical stochastic processes, including standard statistical analysis to obtain mean, mean-square, auto-correlation, covariance, and other parameters [37,38]. Time-domain measurement is a direct method to monitor temporal signal trends, and straightforward analysis is used to construct histograms whose distributions produce the aforementioned parameters of mean, standard deviation, and frequency of occurrence. In addition, when we inquire about how a system signal is related to a signal at other times, we examine correlation functions. An *auto-correlation* function is defined as $\phi_{I}\left(t_{1},t_{2}\right)=\langle I\left(t_{1}\right)I\left(t_{2}\right)\rangle$ or $\phi_{I}\left(\tau \right)=\langle I\left(t\right)I\left(t+\tau \right)\rangle$ for a given current time series I(t) at two different time instances $t_{1}$ and $t_{2}$ or the relative time interval τ, where 〈…〉 indicates an appropriate averaging method from either time or ensemble. A *cross-correlation* function quantifies two different measurement sets denoted by $I_{1}\left(t\right)$ and $I_{2}\left(t\right)$ whose subscript means separate data, written as $\phi_{I_{1}I_{2}}\left(\tau \right)=\langle I_{1}\left(t\right)I_{2}\left(t+\tau \right)\rangle$. While the assessment of time signals provides quantitative information from which insights into the system behavior are gained, it is very hard to confidently isolate a particular noisy signal at a target noise frequency (*f*) underneath strong white background noise in the time domain by differentiating a rapidly oscillating component from a slowly varying one. In such cases, we resort to spectral analysis to investigate measurement data with Fourier transform (FT), which is a standard mathematical tool that converts the time information to its conjugate parameter, frequency. A power spectral density (PSD) is an FT pair of auto- or cross-correlated fluctuating signals, providing the spectral content of the fluctuations in the frequency domain:(1)$$S_{I}\left(\omega \right)=2\int_{-\infty }^{\infty }\phi_{I}\left(\tau \right)e^{-i\omega \tau },$$
(2)$$S_{I_{1}i_{2}}\left(\omega \right)=2\int_{-\infty }^{\infty }\phi_{I_{1}I_{2}}\left(\tau \right)e^{-i\omega \tau }.$$

Figure 2c shows a typical frequency response of PSDs, where three distinct frequency regions are identified: (1) low frequencies dominated by PSDs which are inversely proportional to frequency (red); (2) frequencies with constant (i.e., frequency-independent) PSD, typically called white noise (blue); (3) high frequencies dominated by PSDs which are proportional to frequency (yellow). Noise values remain constant over a certain frequency range, known as white noise, and two examples of frequency-independent PSDs are thermal noise and shot noise. Any electronic system with finite conductance *G* at a temperature *T* exhibits an intrinsic Johnson–Nyquist thermal noise whose current PSD per a given frequency, $S_{I,\mathrm{thermal}}$, has a form of $4k_{B}TG$, where $k_{B}$ is a Boltzmann constant [37]. The explicit presence of *T* in the $S_{I,\mathrm{thermal}}$ formula shows that when *T* decreases, $S_{I,\mathrm{thermal}}$ goes down and eventually reaches zero at *T* = 0 K. Shot noise appears when a net average current $\bar{I}$ flows through the system owing to quantized charge carriers, whose PSD per unit frequency is simply $S_{\bar{I},\mathrm{shot}}=2q\bar{I}$, where *q* is the elementary charge [37]. Another common noise exhibits specific frequency dependence in its PSD: either it grows as frequency increases or it diverges as frequency decreases. The latter especially follows a power-law behavior in frequency as a form of $1/f^{\alpha }$ with a constant α (0.5<α<2.5). It is conventionally called 1/*f* noise or low-frequency noise, and it has several alternative names such as flicker noise or pink (α = 1) or brown (α=2) colored noise.

Low-frequency noise is ubiquitous in nature and is observed in numerous areas of economics, astrophysics, geophysics, biology, neural circuits, engineering, electrical devices, and solid materials. One notable and interesting feature of the 1/f noise is scale invariance, which represents the constant noise power for a fixed decade of the frequency range [39]. Despite several decades of research, the microscopic origins of the 1/f noise are not clear [40]. Most researchers think that these origins would be associated with device structures and materials and the scale invariance would be attributed to fractal phenomena or nonlinear processes [41]. Phenomenological studies of 1/f noise have been conducted to determine the quality of the bulk materials and the interfaces of layered materials. In an electrical circuit, 1/f noise is often the dominant noise source in the low-frequency regions, where PSDs in different electrical parameters hold the following equality, known as Hooge’s relation [42]:(3)$$S_{x,1/f}=A\frac{x^{2}}{f^{\alpha }},\;\mathrm{where}\;x=I,V,\mathrm{or}\;R,$$
with a scaling coefficient *A*. When the relations of I,V, and *R* are linear, the form of the PSD is the same; however, for the nonlinear relations among them, the conversion of the PSD with a selected parameter to another should be carefully investigated. Both the shot noise and the low-frequency PSDs do not have explicit temperature dependencies in their formula that could suggest that the thermal effect would implicitly influence the transport parameter values.

A distinctive low-frequency noise appears as a series of instantaneous switching events in time, which is named burst noise or RTS. Figure 2e plots a simple two-level RTS (bottom) and a complex multi-level RTS (top). According to theoretical and experimental studies, a sharp switching occurs when a certain trapping center in the system either captures or releases a single charge carrier in a random fashion. For instance, short-channel MOSFETs display RTSs in drain currents, where two distinctive currents are observed when a single charge trap independently captures or emits an electron [43,44,45]. As another example, a two-level fluctuator at the superconducting qubit systems is considered to be responsible for the current RTS [46,47]. Each RTS is determined by three quantities of the amplitude between two distinct values of a measurement variable denoted as ΔI and two dwell time constants $\tau_{\mathrm{high}}$ and $\tau_{\mathrm{low}}$ at the high and low levels, as indicated in Figure 2e. Statistically, the trapping and detrapping events obey a simple Poisson process from a series of independent abrupt switches whose distributions are quantified by average characteristic times, ${\bar{\tau }}_{\mathrm{high}}$, and ${\bar{\tau }}_{\mathrm{low}}$ [48], and the corresponding PSD of the Poisson distribution exhibits a Lorentzian spectrum with α = 2. For the RTS in currents, the PSD is expressed as follows:(4)$$S_{I,\mathrm{RTS}}=C\frac{\Delta I^{2}}{1+{\left(2\pi f/f_{c}\right)}^{2}},$$
where *C* is a coefficient and $f_{c}$ is the corner frequency of the RTS spectrum. The Lorentzian RTS PSD is derived from the autocorrelation function of a Markov process according to the Wiener–Khinchin theorem. When we extend a single RTS model into many charge traps in systems resulting from fluctuations in either mobility or charge numbers, the overall system PSD becomes 1/f noise with the power exponent 1, as shown in Figure 2f [49]. Indeed, mathematically, the 1/f PSD is acquired through the summation of many independent Lorentzian RTS PSDs (Figure 2d), where each RTS is associated with a unique defect with particular time constants [50]. Unfortunately, we cannot retrieve the parameters of many traps from the 1/f-PSD lack of information in the frequency domain. Hence, most RTS research focuses only on a single trap with two well-defined states or on the overall 1/f noise, potentially caused by many traps [51]. Recently, as the device sizes shrink in both classical and quantum systems, multi-level RTSs from a few traps occur more frequently and a protocol for their analysis is therefore desired.

### 2.2. Noise Processes in Solid-State Materials

Most solid-state devices comprise different material classes: metals, insulators, semiconductors, and others, in order to benefit from their advantageous electrical, optical, chemical, and physics properties and to ensure their appropriate usage and desirable functionalities. Revolutionary electronic, photonic, optoelectronic, and mechanical systems make the most of intrinsic materials, combinations of different materials, and material dimensions. Recently, dimensional engineering has added a new degree of freedom in stacking up low-dimensional materials to form multitudes of heterostructures, where not only individual materials but also the interfaces of different material stacks play crucial roles in device performance as well as noise processes. At finite temperatures, metals have finite resistance, which is the source of the Johnson–Nyquist noise in equilibrium which can be reduced by either a cool-down or a lower resistance. Even though the purity of materials has been greatly improved, many defects and impurities still exist in bulk semiconductors and oxides. Furthermore, the material’s surface and interface are significantly involved since more charge trap centers or defects tend to exist and they likely interact with charge carriers through the aforementioned capture and release [52,53]. Indeed, the quality of a material is revealed in its noise. The exponent value of the 1/f noise serves as a quantitative measure of the material purity in classical devices and systems. Low-frequency PSDs act as early failure detection to select robust organic light-emitting diodes in mass-production [54], and 1/f noise properties are studied in layered MoTe${}_{2}$ transistors to examine their susceptibility to environmental fluctuations in sensor applications [55]. The 1/f noise and RTS have been measured in a variety of complementary metal-oxide semiconductor (CMOS) devices [56], and recently, as device sizes are shrinking down to the nanoscale, complex RTSs with multiple distinct levels of states occur more frequently [57,58,59]. Table 1 summarizes notable noise trends in several frequency domains where a certain noise source becomes dominant from internal noise sources in solid-state materials.

Similar to classical systems, understanding the origin of noise and controlling noise processes are also primary research problems in the quantum world. An isolated and closed quantum system is stationary but unrealistic and useless since there is no path to pass or retrieve information to and from it. Thus, a practical quantum system is intrinsically open, where qubits are connected to the external environment or the bath. These open quantum systems are no longer time-independent, but their dynamics are tightly affected by the environment. The link to the bath outside is inevitable for information encoding, storage, and retrieval; however, the interactions with the bath also lead to a degradation in coherence and loss of information through dissipation and dephasing processes because the coherence of quantum states is fragile and vulnerable to various fluctuations and disturbances in the environment. Therefore, time-fluctuating signals almost always appear, and noise-susceptible decoherence is observed via the interplay of bath relaxation time and system dynamics [60,61]. In addition, imperfect gate operations on a single qubit or arrays of qubits and limitations of the measurement apparatus can also add noise. The relaxation time $T_{1}$ and the dephasing time $T_{2}$ are measured as figures of merit to quantify how noisy quantum systems are. When ensembles of qubits are involved, their dephasing time is symbolized as $T_{2}^{*}$, which is typically shorter than $T_{2}$. QIP also considers an opportunity to exploit noise collectively to enhance the performance analogous to stochastic resonance or differential privacy in the classical world. One attempt was made to tackle a machine-learning classification problem where quantum data are protected from adversarial attacks by quantum depolarization noise in a quantum circuit [62]. This area will continue to attract more attention because making unavoidable noise processes allied to quantum coherence is no doubt favorable for large-scale, fault-tolerant quantum processors.

Numerous studies have been conducted to elucidate the origins of fluctuations and to identify how to mitigate them in quantum processors [63]. Efforts in this area ultimately advance error correction and mitigation strategies in digital computation [64,65]. It is natural that the properties and behavior of noise are highly idiosyncratic in each quantum processor architecture, where both classical and quantum fluctuations coexist under Gaussian and non-Gaussian statistics [66]. Low-frequency behaviors aggravating decoherence are clearly observed in the form of RTSs associated with two-level or quantum fluctuators in materials [46,47], and the noise spectral density in qubit parameters exhibits a power-law trend [46,66,67]. The next subsection is devoted to explaining our step-by-step analysis of RTS time tracing method adopting neural network models, which is essential to quantify RTS parameters in complicated signals.

### 2.3. Random Telegraph Signal Analysis

Superconducting qubit systems and semiconductor spin qubit systems find single-trap and multi-trap RTSs in their transport measurements, as discussed in the previous Section 2 and Section 3. A common source of RTS is a defect to be modeled as a two-level fluctuator or a charge trap, often at material interfaces. The analysis of a single-trap RTS results in amplitude between two levels and average dwell times at the high and the low levels. Recently, for miniaturized devices down to nanometer sizes, complex RTSs with multiple distinct levels of states are frequently encountered, possibly induced from only a few countable traps [57,68,69,70]. Thus, a tool to assess multi-level RTS signals is required [69,71]. However, to our best knowledge, there are only a limited number of references providing systematic analyses of a few-trap RTS—an intermediary regime between a single-trap RTS and multi-trap 1/f noise. To isolate individual RTSs to obtain three parameters is a pre-requisite prior to the discussion of the overall device’s performance. A major challenge to obtaining accurate information about each RTS is the existence of background noise. Our group develops a systematic numerical protocol for single-trap and few-trap RTSs in three major steps: to identify the presence of the RTS in noisy measurement data, to decompose complex RTS components into sub-component RTSs associated with each trap, and to extract quantitative parameters for each RTS. We aim to collect quantitative information on each RTS in terms of a modulation amplitude ($\Delta X_{\mathrm{RTS}}$) from the first two steps, where *X* represents a parameter chosen in experiments, and the characteristic capture (${\bar{\tau }}_{c}$) and emission (${\bar{\tau }}_{e}$) times of individual RTS in the last step.

*Step 1: Density Estimation.* First, the RTS states are identified and classified from raw noisy data in terms of the trap number ($N_{\mathrm{trap}}$) and the background noise size, which become seed estimates for the subsequent generation of the RTS model. To minimize other fluctuations such as white noise [72] in the raw data, a moving average (MA) allows us to create a density distribution of the filtered data using kernel density estimation (KDE). Figure 3a compares the kernel density distributions before (gray) and after (blue) filtering beyond simple histograms, where MA effectively removes background noise. The advantage of KDE lies in the fact that a continuously differentiable kernel function leads to a smoother density distribution in contrast to a discrete histogram. Together with a Gaussian kernel, the number of false peaks caused by fluctuating white noise is reduced while the ability to resolve close peaks remains the same. The number of peaks is related to $N_{\mathrm{trap}}$, and at maximum, $2^{N_{\mathrm{trap}}}$ peaks are present if all traps are mutually exclusive and independent. In practice, complex RTS has smaller peaks, partly due to strong background noise to hinder all peaks from being distinctively visible or partly due to missing levels from internal correlation and many other reasons.

*Step 2: Decomposition.* Once all pronounced peaks are recognized, we can represent the density of a simple RTS component as the mixture model of two Gaussian functions, $G_{\mathrm{high}}$ and $G_{\mathrm{low}}$. Since independent traps naturally indicate that the corresponding RTS components are also independent, the total probability distribution $G_{T}\left(x\right)$ from the multi-trap RTS is constructed via the convolution of individual probability functions, where *x* symbolizes a measurement variable [73]. Figure 3b presents the results of the decomposition for a three-trap RTS example, from which $\Delta_{\mathrm{RTS}}$ of each component is evaluated.

*Step 3: Digitization.* The final step is to obtain digitized signals to find characteristic dwell times at each RTS level. We recognize RTS as an example of a sequence of labeling tasks with the same input and output length. Elaborately at a time incidence $t_{i}$, an input datum $x_{i}$ is assigned to have a specific output $y_{i}$ between two states. True RTS is a Markovian process without any memory, but the knowledge of the previous and future values can help to accurately assign the output in RTS. In order to predict an output at $t_{i}$ with higher confidence, the model considers outputs at incidents that are one, two, or *n* steps earlier (i.e., $t_{i-1}$, $t_{i-2}$, or $t_{i-n}$ (past)), as well as incidents that are one, two, or *n* steps later (i.e., $t_{i+1}$, $t_{i+2}$ or $t_{i+n}$ (future)). This insight motivates us to set up bi-layer deep neural network models and train them with past and future data, as shown in the diagram (Figure 3c).

The validation of our multi-level RTS analysis protocol was carried out with synthesized 720 RTS data by varying $N_{\mathrm{trap}}$, background noise types (Gaussian white and pink noise), the size of the background noise amplitudes, and the kinds of anomalous RTSs [74]. The confidence and the limitation of our algorithms to handle multi-level RTS data are gained from a statistical error analysis in comparison with the obtained RTS parameter values from this protocol and the ground truth design values in synthesis. We learn that reasonably clean two-level RTS data can be quickly assessed by the hidden Markov model (HMM) with high accuracy, but the HMM cannot be straightforwardly applied to multi-level RTS, where this systematic algorithm can contribute a lot. In addition, our protocol utilizes open-source codes and takes the necessary information from previous steps; therefore, it is agnostic and can investigate any multi-level RTS data in various fields.

Commonly, we admit that a big factor limiting quantum coherence is also the quality of materials with which qubits and their environments are built [31]. The next two sections survey material-inherent noise processes in two solid-state quantum architectures based on superconducting qubits (Section 3) and semiconductor spin qubits (Section 4) that have shown remarkable progress toward large-scale quantum processors through the integration of more qubits and efficient classical control circuits [75]. However, these two architectures have lots of room to improve their noisy processes for the construction of quantum superior processors beyond the NISQ systems. Section 5 summarizes the impact of solid-state materials in surface trapped-ion systems, which are the future of large-scale, ion-based quantum processors, in the context of noise properties. These sections aim to review notable research activities of investigating noise properties and learning the important lessons for smarter noise control maneuvering.

## 3. Superconducting Qubit Quantum Processors

Superconducting quantum systems are one of the most advanced solid-state platforms for the construction of commercial quantum hardware at present. These quantum systems are capable of performing quantum computation and quantum simulation close to the stage needed to accomplish fault-tolerant operations and quantum supremacy [17,76,77,78]. Superconducting qubits are defined by their assorted degrees of freedom: charge [79,80], phase [81], and flux [82,83]. At the heart of it all is the Josephson junction (JJ) of a superconductor–insulator–superconductor structure, a diagram of which is given in Figure 4a. Together with other circuit elements (e.g., a capacitor or an inductor), a superconducting loop with single or multiple JJs can form unique quantum circuits where |0〉 and |1〉 states are specified [84]. In the JJ tunnel junction, many things happen: electrons or Cooper pairs tunnel through a thin insulator at a normal flow or in persistent supercurrents and a magnetic flux penetrates through the superconducting loop. For unrivaled quantum information processors with superconducting qubits, long coherence times are desirable and required for a series of single- and two-qubit operations with high fidelity. In reality, individual superconducting qubits and qubit arrays suffer from fluctuations in charge [85,86], flux [87,88,89], or junction currents, as well as dielectric loss [90], which would limit coherence times as a roadblock towards large-scale quantum computers [91]. Elucidating the exact microscopic origins of various fluctuations has become an active research topic in order to enhance superconducting quantum system coherence. Three notable strategic approaches have been pursued to defeat the noise problems. First, certain parts of the circuits are revised to bypass the issues and for reduced noise. Second, novel fabrication processes have been developed and attempted for improved quality. Third, new materials to overcome the aforementioned material limitations are actively being searched for and explored [92]. Some examples in these directions are discussed below.

Accumulated experimental research tells us that unsettled fluctuations are often closely pertinent to associated conducting and insulating materials in bulk, surface, and interfaces between different materials [93]. The chief materials used to make the majority of JJs are aluminum and niobium, alongside their oxides. Early JJs employed amorphous aluminum oxides, and the vertical tunnel junctions were fabricated through a double shadow-evaporation technique [94]. One onerous trouble of amorphous tunnel barriers is their two-level fluctuators (TLFs), which are recognized as spectral splittings in qubit spectroscopy [90,95]. TLFs can interact with qubits, whose energy and phase become disturbed, consequently shortening $T_{1}$ and $T_{2}$ and exhibiting low-frequency 1/f noise. As a remedy, Oh et al. grew a single-crystal tunnel barrier with an epitaxy method for flux-biased phase qubits. Figure 4b presents the RHEED image at each growth step, where bright periodic peaks confirm crystalline materials. Indeed, the features of splittings are strongly suppressed in the epitaxial barrier case, in which a highly controllable growth method can produce a high-quality crystalline insulator with a smaller number of TLFs in contrast to the amorphous-barrier counterpart.

Early on, phase qubits displayed strong, low-frequency 1/f noise [96], and they suffered from much shorter coherence time than charge and flux qubits. A dominant source of fast decoherence in phase qubits was attributed to dielectric loss from discrete defects, modeled as TLFs in bulk insulating materials and tunnel junctions [90]. This decoherence issue was mitigated by smaller phase qubit designs where the density of the TLS is lower in a given footprint. Better insulating materials also helped to increase coherence times by 20 times [90]. Recently, Osman et al. demonstrated a one-step fabrication technique, the so-called patch-integrated cross-type (PICT) method [97]. Since it can make a junction and a patch layer at once, the process is simple and quick. The authors fabricated more than 3600 aluminum-based junctions in this method and the JJ variation is 2.5–6.3% on the wafer scale. Transmon qubits are prepared by PICT to measure their qubit parameters, yielding $T_{1}$∼ 50–60 μs, which is comparable to the transmons made by a previous method. This is an encouraging result that is promising for scalable and reproducible JJs.

**Figure 4 materials-16-02561-f004:**
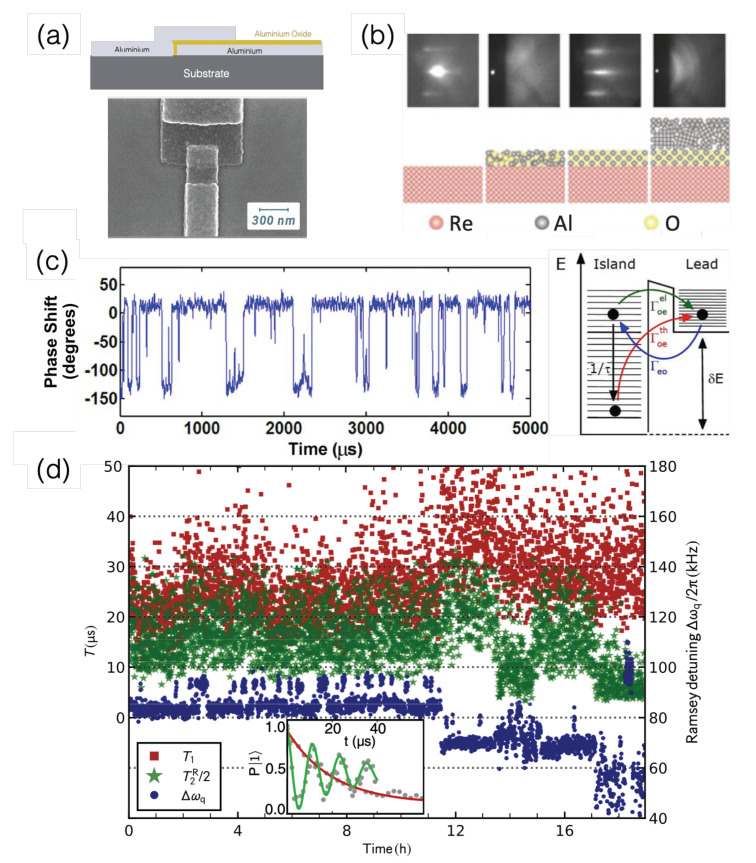
(**a**) A cross-section (top) and a scanning electron microscope (SEM) image (bottom) of a simple Josephson junction (JJ) based on aluminum material. Images from https://angstromengineering.com/josephson-junction-thin-film-deposition-superconducting-circuits/, accessed on 20 August 2022 (public domain). (**b**) RHEED images (top) at the material growth sequences to form a single-crystal Al2O3 tunnel barrier during an epitaxial growth on epitaxial rhenium (Re) layer, adapted with permission from Ref. [98]; ©2006 American Physical Society. (**c**) Random telegraph signals (left) and a diagram of quasiparticle tunneling in a charge qubit, where $\Gamma_{\mathrm{eo}}$ is the tunneling rate from a lead to the island; the tunneling rate from the island to the lead by thermal excitation and by elastic interaction between quasi-particles are denoted as $\Gamma_{\mathrm{oe}}^{\mathrm{el}}$ and $\Gamma_{\mathrm{oe}}^{\mathrm{th}}$, respectively (right). The images are adapted with permission from Ref. [46]; ©2008 American Physical Society. (**d**) Time traces of JJ parameters in a transmon qubit. $T_{1}$ and $T_{2}$ are taken from the Ramsey detuning fit in frequency change $\Delta \omega_{q}$ with the probability of the excited state population presented in the inset adapted with permission fromRef. [99]; ©2019 American Physical Society.

Instantaneous switching signals in temporal measurements are seen in superconducting charge qubits, which consist of a superconducting island and a JJ. Known as a Cooper pair box, the qubit states rely on the Cooper pair tunneling. Unfortunately, quasiparticles other than Cooper pairs can also tunnel into islands [86,100], which are regarded as undesirable occurrences and noise processes. Figure 4c shows a time trace of the phase shift in the reflected radio-frequency signals applied to two charge qubits [46]. Each phase jump event represents the trapping or detrapping of an individual quasiparticle into a Coulomb island from a superconducting lead. A kinetic theory models the quasiparticle movements between the island and the lead, the energy scales of which are sketched in the right of Figure 4c. The dwell times of RTSs determine the quasiparticle tunneling rates, and they are temperature-dependent. Since these quasiparticle tunnelings are notoriously limited a short coherence time, researchers have attempted to modify the superconducting gap profile spatially across the junctions to allow only Cooper pair tunnelings as much as possible. In materials science, either oxygen doping [100] or film thickness engineering [101] is exploited to block quasiparticle tunnelings.

Fluctuations in charge numbers also induce decoherence in charge qubits [85,86]. A strategy to subdue these fluctuations is to design advanced charge qubits called transmons by shunting their transmission line [85,86]. The large capacitance of transmons reduces charging energy needed to put the qubit in a different operation regime where the junction energy is stronger. This elongates coherence time $T_{2}$ to be ∼3 μs. Figure 4d monitors the 19-h tracking of transmon parameters in the time domain [99]. The qubit frequency has multi-level RTS-like shifts (blue), and both $T_{1}$ and Ramsey dephasing time $T_{2}^{R}$ have wild variations. These measurements are exemplary to address how the qubit frequency fluctuations affect $T_{2}$. The RTSs are induced by TLFs and depend on the TLF types. The correlations between the qubit frequency shift and $T_{2}$ are positive on the time scale of seconds but negative on the order of days, which may be interpreted as a single dominant TLF near the conductor edge.

Lastly, Josephson persistent-current flux qubits stand on an integral quantum unit of flux $\Phi_{0}=h/2e$, where *h* is a Planck constant and the Cooper pair charge is 2e through a superconducting loop. As the equation implies, flux fluctuations can be caused by charge variations. Flux qubits also encounter similar noise issues of resistive loss to produce charge noise and 1/f flux noise, impeding qubit coherence metrics of $T_{1}$ and $T_{2}$. Over the last decade, several approaches have been made to elongate $T_{1}$ and $T_{2}$ by symmetric device design through an optimal bias condition [102,103,104], a capacitive shunting scheme [105], and a bulk cavity [106] or coplanar waveguides [107]. Typically, the 1/f flux noise is engendered by TLS to couple with qubits at the interface [87,92,96,108]. The research group led by Dr. William D. Oliver recognized the non-Gaussian nature of noise when a qubit is in a strong coupling regime with the environment or in a nonlinear energy dispersion where Gaussian approximation is no longer valid [109].

Through active theoretical and experimental research efforts, the superconducting qubit community has accrued tremendous knowledge and strategies to improve the fifth-order magnitude from 1 ns to 0.3 ms in $T_{1}$ and from 2 ns to >100 μs [110,111] by exploiting the smart design of the quantum circuits, state-of-the-art fabrication techniques, and new materials [112]. Upon this remarkable progress, researchers have laid out a manufacturing blueprint to demonstrate superconducting qubit circuits with the help of mature industry-scale CMOS facilities toward practical large-scale superconducting quantum architectures [113].

## 4. Solid-State Spin-Qubit Quantum Systems

Spin qubit systems employ spin-up and spin-down states of electrons or holes in solid-state materials as a celebrated two-level basis for quantum information processing in computation, communication, simulation, and sensing. This spin-based quantum architecture was conceived by Daniel Loss and David DiVincenzo in the seminal paper published in 1998 [23]. A gate-defined quantum dot (QD) hosts a charge in an island whose energy states are quantized, owing to its strong spatial confinement. Figure 5a illustrates a single QD whose region is determined by several gate electrodes. This quantum structure was successfully made of GaAs, Si, and Si/Ge semiconductors. Electrical transport properties have been thoroughly characterized, and the beautiful Coulomb blockade phenomena became known to the world [114]. Zero-dimensional QDs are called “artificial atoms”, where quantized energy levels are formed by intentionally tailored potentials in semiconductors as a result of a quantum-confined effect in all dimensions.

Semiconductor QDs are conveniently fabricated using existing CMOS (complementary metal oxide semiconductor) facilities and state-of-the-art nanofabrication technologies. A charge qubit is built upon the occupation of electrons under confining potentials. It has also been demonstrated in the QD system [119] that a single spin qubit is defined by a magnetic field to split the spin-qubit states. The spins are controlled via electron spin resonance (ESR) [120] or electric dipole spin resonance (EDSR) [121]. Advanced qubit systems have been concocted: for instance, a singlet-triplet qubit, which is a kind of mixed charge and spin qubit [122], or a hybrid qubit, where three electrons are involved without magnetic fields [123]. These novel qubit systems are possible with mature materials and advanced processing technologies; however, they still exhibit technical limitations, requiring cryogenic environments and suffering from drawbacks such as a short decoherence time and weak entanglement.

Another relatively new and spotlighted spin-qubit platform is the color centers derived from specific defects in materials [124]. The conventional construction of a color center is a nitrogen vacancy, which acts as a point defect in diamonds and is illustrated in Figure 5b. They have attractive qubit operating conditions in terms of temperatures and coherence times. The color-centers can act at room temperature [125], and their relatively long coherence times (∼μs) accommodate ultrafast (ps or fs) gate operation compared to QD subits of the III-V semiconductors that have reasonable coherence times (∼μs) with slow microwave (μs) control [126].

Semiconductors are the host for both QDs and color center qubits, where spin states of electrons and holes are rather brittle and greatly affected by surrounding nuclear spins in the host, which is a major root cause of the noise in QDs responsible for their shorter coherence time. Another noise source in QDs is the spin fluctuations from trapped carriers in the oxide interface, the non-uniform interface, and unstable potentials. On the other hand, the main root cause of the noise in the color centers is the unavoidable lattice defects of the host material during fabrication [127], as well as other unwanted imperfections such as poor-quality surfaces and dangling bonds as sources of magnetic noise [128]. GaAs and AlGaAs are natural choices for the construction of QD spin-qubit platforms by virtue of their high mobility as a manifestation of material quality, which is certainly favorable for minimizing charge noise. In fact, the first coplanar double-QD system to realize spin qubits was demonstrated with coherent Rabi oscillations on GaAs two-dimensional electron gas (2DEG) systems in 2006 [129]. However, this material has a fatal obstacle of a short μs-decoherence time, primarily associated with nuclear spin noise in GaAs-based III-V semiconductors. Notwithstanding strong efforts were made to accommodate the material-inherent noise with possible solutions of advanced dynamical decoupling and fast Hamiltonian estimation, promising alternative materials have been searched for enthusiastically.

A versatile and mature semiconductor is Si. As an excellent host material, it has abundant isotopes, among which the three most dominant stable isotopes in natural Si are ${}^{28}$Si (92.23%), ${}^{29}$Si (4.67%), and ${}^{30}$Si (3.1%). While ${}^{28}$Si and ${}^{30}$Si are nuclear-spin-free and desirable for quantum technologies, ∼5% ${}^{29}$Si forms a nuclear spin bath as a source of magnetic noise to fluctuate the spins of electrons or holes. With the help of a cutting-edge isotope engineering technique to remove ${}^{29}$Si, highly purified and isotopically enhanced ${}^{28}$Si can make Si devices including Si/SiGe, Si-MOS, and CMOS structures [130]. Indeed, a ${}^{28}$Si Avogadro crystal features a significantly long coherence time of seconds with donors at cryogenic temperatures [131] or 39 min with ionized donors even at room temperature [132]. Another approach to handle the magnetic noise issue is to utilize hole spins instead of electron ones because holes are less vulnerable to nuclear spin interactions because of their anisotropic *p*-orbital symmetry and large spin–orbit coupling in Ge [133] and Si-MOS [134]. Furthermore, among carbon families, carbon nanotubes [135,136] and graphene are good host material candidates with small nuclear spin densities; however, graphene is more attractive as it has weak interactions between the spins and orbitals of electrons [137,138]. Diamond has been the main host material for color center spin qubits as a consequence of its large bandgap energy over a long coherence time with nitrogen vacancy (NV) and silicon vacancy (SiV). SiC is also actively being investigated since it can operate at room temperature as a color center material [139].

A typical microwave control between the QD spin states is in operation at the GHz range, but 1/f noise puts a ceiling on the qubit decoherence and spin dephasing time ($T_{2}^{*}$). There are two types of noise processes: electrical charge noise and magnetic spin noise. The Tarucha group demonstrated Si/Ge QD qubits monitored by a single electron transistor (SET) sensor with >99.9% single-qubit gate fidelity and 20 μs-long phase coherence time [140]. A primary factor of enhancement is to exploit an isotopically purified Si to measure the $1/f^{\alpha }$-PSD, from which they concluded that the qubit decoherence is dominantly limited by charge fluctuations. Both the Ramsey and advanced Carr–Purcell–Meiboom–Gill dynamical decoupling schemes yield a PSD which follows along the $1/f^{\alpha }$-power law for almost eight decades. The qubit energy splitting noise is closely related to the microwave detuning frequency ($\Delta f_{\mathrm{MW}}$) and is converted to the PSD in the frequency domain. The noise trends qualitatively account for the $T_{2}^{*}$, which is estimated by the PSD power law:(5)$$T_{2}^{*}\left(t_{m}\right)={\left(\frac{4\pi^{2}S_{0}}{\alpha -1}\left(t_{m}^{\left(\alpha -1\right)}-t_{e}^{\left(\alpha -1\right)}\right)\right)}^{-\frac{1}{2}},$$
where α and $S_{0}$ are the exponent and coefficient of the PSD, respectively; $t_{m}$ is the total measurement time; and $t_{e}$ is the evolution time. The PSD from the current noise detected by the SET sensor is in accordance with the PSD from the splitting noise [117] in Figure 5e, where the nucleus spin noise is modeled and examined in the low-frequency regime as $1/f^{2}$ [141] and a high magnetic field tends to lower the PSD [142].

Similar to the QDs, the color centers are also analyzed by low-frequency noise PSDs in terms of electric and magnetic field noises. Since one for of electric field noise is hypothetically due to fluctuations in the surface charge, Chrostoski et al. coated six different protective surface layers and examined their impact on electric noise PSD. Indeed, distinctly different PSD trends are observed in Figure 5f with regard to the four material parameters of the frequency-dependent electric permittivity, Young’s modulus, the material’s characteristic relaxation time, and a broadening factor in the dielectric function. Specifically, solid materials are good at the high-frequency ranges and a smaller exponent of the power law is generally observed in the liquid materials [118]. The flat region at the low-frequency regime in the PSD is attributed to dipole–dipole interactions, and the magnetic field noise is studied with a view of the surface and bulk impurity interactions based on the PSDs [143]. Overall, the spin qubits from the QD and color center platforms undergo various noise sources from the noises on the surface or in bulk materials, which are easily detected and examined with PSDs in the low-frequency regime while the systems are controlled by high-frequency microwaves or lasers. Indeed, this PSD analysis method is a reliable monitoring tool to provide technical insights and possible solutions that can handle noise problems in a more efficient and consistent manner.

## 5. Surface Trapped-Ion Quantum Systems

In this section, we turn to trapped-ion quantum systems, another successful platform for rendering quantum technologies and realizing commercial quantum computers with high fidelity of single-qubit and two-qubit gates [18,19]. Basically, charged ions in an ultrahigh vacuum chamber are trapped in a confining potential with a variety of trap geometries. A linear Paul trap is created by two end-cap electrodes of a static electric field and four electrodes in a quadrupole configuration, where radio-frequency electrical fields are applied [144,145]. A cylindrical-shaped electrode stack forms a Penning trap in three dimensions to capture myriads of ions inside in the presence of both axial magnetic and quadrupole electric fields [146]. These systems are often placed inside a big vacuum chamber, whose bulky footprint becomes a hindrance to large-scale trapped-ion quantum systems. To overcome this challenge of miniaturization, researchers have resorted to a solution to adapt solid-state microfabrication techniques to making surface ion traps [147,148,149]; using such techniques, an integrated architecture of many trapped ions looks feasible. Marked progress has been made recently, and surface ion trap systems can now be combined with a waveguide, which can allow an individual ion to be addressed by photons [150]. This certainly adds a favorable trait to ion-photonic quantum circuits.

Trapped-ion systems enjoy long-lived electronic energy states of atomic ions, where quantum information is encoded. The electric fields create a trapping potential, and two energy states are selected to define |0〉 and |1〉. The transition between two states can be governed by a laser for optical quadrupole transition or by microwave. They have desirable features such as an impressively long coherence time of up to 1 h for a single-qubit system [151]. A record-high single-qubit fidelity of >99.995% was reported under laser control [152,153] and of 99.9998% under microwave control [154]. A remarkable two-qubit fidelity of 99.9% is possible using the optical method [153,155]. Despite such impressive performance, in order to construct fault-tolerant, trapped-ion quantum architecture, there is still a strong need to understand system noise sources and decoherence limiting factors, which depend on actual trap structures. Owing to the basic trapping principle of charged ions, electric field noise produced by the system’s interaction with the environment plays a significant role in the motion of ions and produces spurious effects, such as the transition between ionic vibrational states [147,156]. Fluctuating electric fields in the order of MHz heat ions by transitioning them to higher-energy modes, which can influence other ions as well. Since stable ion positions are essential to achieve an accurate and sustainable control of gate operations, noise sources to drift ion positions should be suppressed.

Integrated large-scale trapped-ion quantum systems are sought after as the ultimate goal for quantum-advantageous computation, communications, simulations, and sensing by harnessing optical and electrical technologies for the initialization and operation of qubits and measurements and the storage of information with the substantial support of materials science. To accomplish this goal, microfabricated surface traps have been developed by combining different classes of materials. Figure 6a illustrates a trap chip mounted in a carrier (Figure 6a, left), and an optical microscope image (Figure 6a, right) shows a real surface trap on a substrate, which consists of metallic DC and RF electrodes and dielectrics to separate electrodes. The fabricated surface is bumpy at the atomic scale due to the grains of the elements, as observed in a scanning electron microscope image (Figure 6a, middle). Owing to the fact that a non-uniform microscopic surface generates an unwanted effect, the same noise issues of electric field fluctuations and ion drift motion also exist on surfaces or interfaces between different materials [157].

Figure 6b portrays plausible scenarios of noises in surface electrodes, including Johnson noise due to resistance at finite temperature (top), diffusion from inhomogeneous work function profiles given adsorbates on a metal surface (middle), or dipole fluctuations of adsorbed atoms (bottom) [158]. The electric field noise PSD, $S_{E}$, is a standard measure to assess the trap quality and its frequency response $S_{E}\left(f\right)$ can also differentiate the dominance of the aforementioned noise sources given a situation by looking at the exponent α of the $1/f^{\alpha }$ response, for example, α=2 for the Johnson noise, α=3/2 for local work function variations, and α=0−2 for dipole fluctuations. Experimentally, $S_{E}$ can be inferred from the ion heating rates as a function of ion-surface distances *d*. The heating rates are high when the ions are closer to the surface, exhibiting a distance scaling behavior of $d^{-\beta }$ shown in Figure 6c [159]. The Häffner group reported a smaller value of β = 2.6 (blue squares in Figure 6c) than the previous observation of β=4 [156,160,161], which may suggest to consider spatial noise correlations in a planar electrode geometry.

**Figure 6 materials-16-02561-f006:**
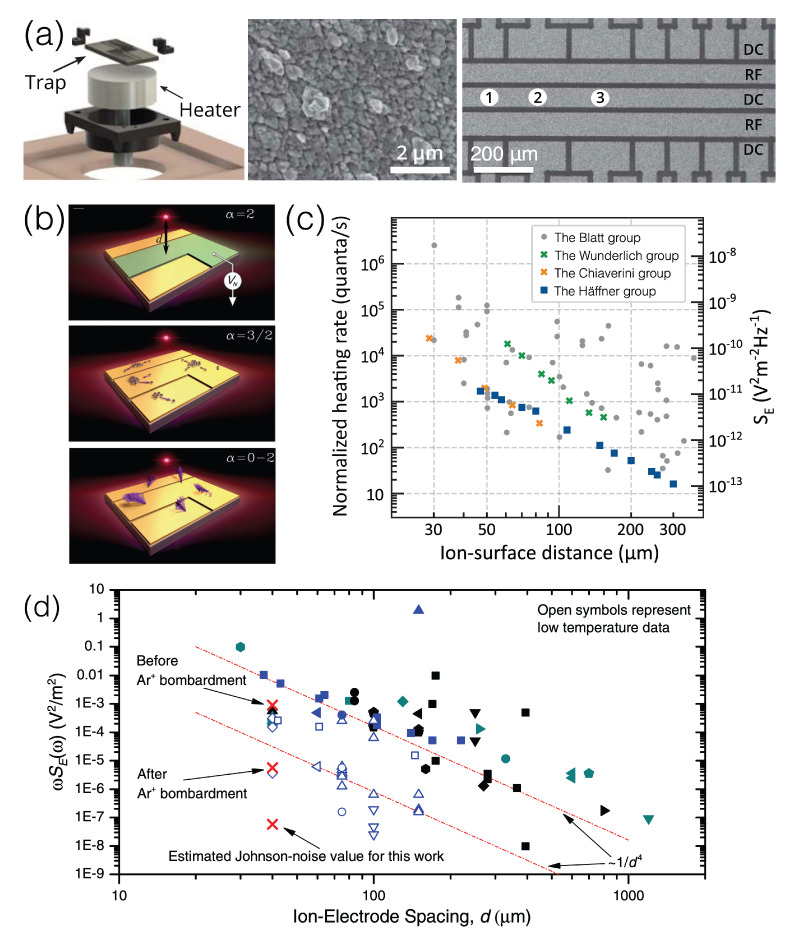
(**a**) A schematic of a surface ion trap with DC and RF electrodes and a scanning electron microscope image of the Al and Cu surface where a Ca${}^{+}$ ion is trapped above 72 μm, reproduced with permission from Ref. [147]; ©2019 American Physical Society. (**b**) Three scenarios of surface noise sources: Johnson noise (top), local work function variations of metals due to adsorbates (middle), and dipole drift (bottom), adapted with permission from Ref. [158]; ©2013 Springer Nature. (**c**) Distance-scaling behavior of heating rates as a function of the ion–surface distance *d*. Gray circles are for single-ion motional heating, but blue squares follow the $d^{-2.6}$ trend, adapted with permission from Ref. [159]; ©2019 American Physical Society. (**d**) In situ argon-ion cleaning to reduce electric field noise. Image is adapted with permission from Ref. [162]; ©2012 American Physical Society.

When an electric field is applied to metal electrodes, the non-zero resistance of metal makes the electric field fluctuate, consequently heating up the ions and changing the ions’ motional states. While the origins of these heating mechanisms are not clearly identified, the surface condition plays a crucial role to reduce heating rates. Since the normalized $S_{E}$ of the surface ion traps is often 10,000 higher than the Johnson–Nyquist noise PSD (red x symbol, Figure 6d), Dr. Wineland’s group attempted an argon-ion-beam cleaning treatment to remove surface adsorbates, which indeed reduced the normalized electric field noise PSD ($\omega S_{E}\left(\omega \right)$) in angular frequency ω by 100 times, as seen in Figure 6d, emphasizing the importance of the electrode surface quality [162]. Furthermore, in principle, the resistive loss in metal electrodes can be removed by using superconductors whose resistance is 0 below a critical phase transition temperature. A couple of research groups have fabricated high-temperature superconducting ion traps and scrutinized their electric field noise carefully [163,164]. While the resistance-induced electric field noise is suppressed in superconducting materials such as niobium [160] or high-temperature superconductors of yttrium barium copper oxide (YBCO) [163], there are still remaining noise sources on their surfaces.

Recently, Ivory et al. made an integrated surface ion trap with a dielectric waveguide in Si fabrication technologies and studied the impact of dielectric materials on heating rates. While a photo-induced charging effect in a dielectric layer is a concern, the heating rate remains at the same level as that with ions only at the closest distance to the surface [150]. This work gives an auspicious sign to confirm that dielectric-based photonic structures may not necessarily worsen noise in support of scalable ion-trap quantum processors with microfabrication technologies.

## 6. Conclusions

We are in the middle of the Quantum 2.0 era, whose technologies impact computation, communication, simulation, and sensing and whose disruptive potential and superiority arise from the fundamental principles of quantum superposition for infinite parallelism and entanglement to achieve ultimate correlation. In the last two decades, remarkable advancements in building quantum information processors have been achieved, and mid-scale commercial quantum computers on several prominent material systems, such as superconducting, trapped-ion, semiconductor spin, and photon qubit systems, are available via cloud services provided by global corporations and start-up companies. However, the fidelity of quantum gate operations and coherence metrics are still not high enough to realize fault-tolerant quantum processors, a situation represented by the coined acronym of NISQ. Each platform suffers from several noise processes associated with materials and their structures to initiate faulty operations and aggravate decoherence. Often, low-frequency noise properties reveal the purity of materials and quality of interfaces, where individual particle tunnelings would create temporal fluctuations.

This review surveys some active noise studies and strong efforts to understand the origins of fluctuations and to mitigate noise in three material platforms with superconducting qubits, semiconductor spin qubits, and trapped-ion quantum systems. In order to avoid or subdue unwanted fluctuations, strategic attempts are placed in designing smarter structures, fabricating high-purity samples with cleaner and more advanced methods, and searching or synthesizing new promising materials. We further notice that random telegraph signals, as an important specific form of low-frequency noise, appear in most quantum systems, and the systematic analysis protocol for these signals is introduced to quantify relevant information of trapping centers, which influence single-particle dynamics. Table 2 summarizes four qubit platforms where solid-state materials are used to construct quantum information processors. Since each platform suffers from different types of noise and their origins, the table captures some reported values of characteristic times ($T_{1}$, $T_{2}$) and gate fidelities ($F_{1}$ for a single-qubit gate and $F_{2}$ for a two-qubit gate) as a possible representative of figure of merit. However, one thing we would like to emphasize is that ranking these platforms from the quantities in Table 2 should be avoided because they do not reveal the complete performance measures of each platform owing to their own unique challenges and advantages. There are far more research endeavors beyond the content of this review which have overcome impediments to achieving large-scale quantum advantageous computation and communication architectures with controllable error corrections in the near future. It is an exciting time to be witness the journey to a better world supported by revolutionary quantum technologies.

## Figures and Tables

**Figure 1 materials-16-02561-f001:**
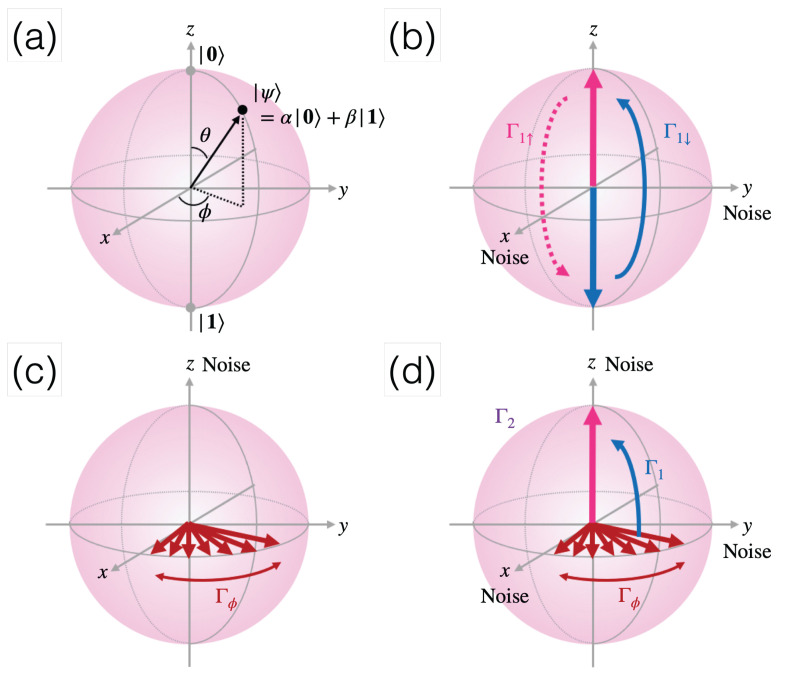
(**a**) A Bloch sphere to locate a qubit state: the ground state and the excited state are represented along the *z*-axis as |0〉 and |1〉 at the north and south poles, respectively. The *z*-axis acts as a longitudinal direction, whose transverse axes are along the *x* and *y* directions. (**b**) Longitudinal relaxation rates ($\Gamma_{1s}$, where s= ↑ or ↓) induced by the energy exchange between two spin states caused by transverse noise sources in the x−y plane. (**c**) A pure dephasing rate ($\Gamma_{\phi }$) in the transverse plane due to adiabatic perturbations along the *z* axis. Adiabatic perturbation means that a system evolves according to slow degrees of freedom. (**d**) Transverse relaxation rate ($\Gamma_{2}$) from a combination of $\Gamma_{1}/2$ and $\Gamma_{\phi }$. The above images are reconstructed from Figure 4 in Ref. [2] with the permission of AIP Publishing; ©2019 AIP Publishing.

**Figure 2 materials-16-02561-f002:**
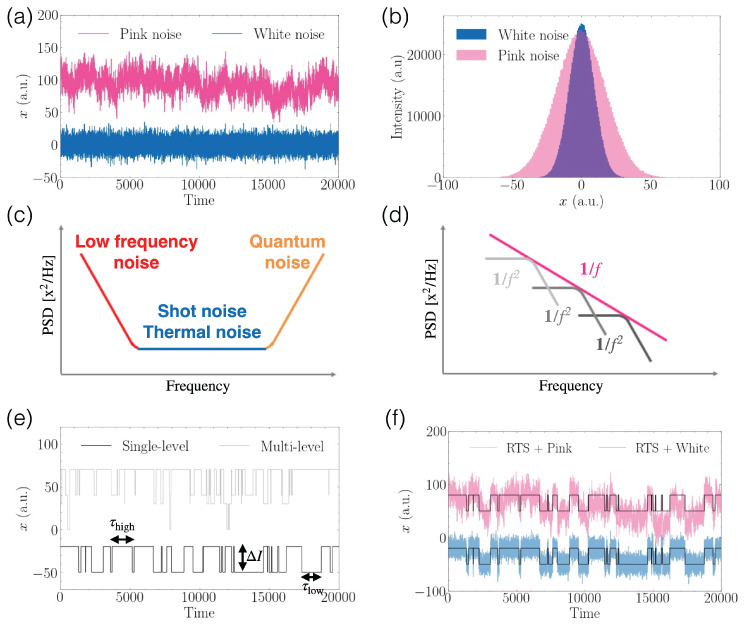
(**a**) Gaussian pink noise (top, pink) and white noise (bottom, blue) of a signal *x* in the time domain and their histograms (**b**). (**c**) Three kinds of noises by power spectral density (PSD) of parameter *x* in the frequency domain. (**d**) 1/f and 1/$f^{2}$ noises in the frequency domain; 1/f noise is thought of as the summing of $1/f^{2}$ noises. (**e**) Representative single-level (bottom) and multi-level (top) random telegraph signals (RTSs) in the time domain. (**f**) A single-level RTS mixed with pink noise (top, pink) and white noise (bottom, blue).

**Figure 3 materials-16-02561-f003:**
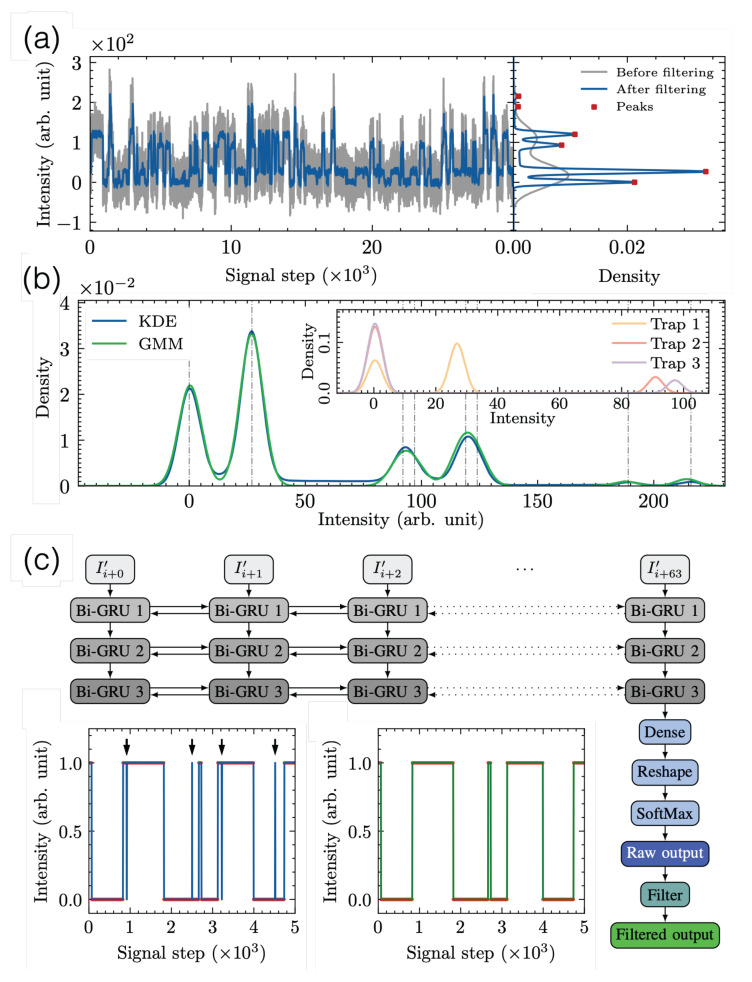
(**a**) (Left) A time-trace segment of a two-trap random telegraph signal with Gaussian background white noise with before (gray) and after (blue) moving average filtering in Step 1. (Right) The continuous kernel density plots are given with much sharper peak distributions for the filtered signals. (**b**) In Step 2, individual traps are decomposed from the kernel density estimate (KDE, pink) and Gaussian mixture model (GMM, green). Both results are consistent in this case since the white noise is not severe. (**c**) The structure of a bidirectional recurrent neural network in Step 3, where $I^{\prime }$ is the raw intensity signal (*I*) normalized to between zero and one, 0 to 63 is the index within the batch, and *i* is the batch’s offset from the first value in the signal. (Inset) Digitization errors are false switching events due to background noise indicated by arrows (left) in comparison to the synthesized ground-truth data (right).

**Figure 5 materials-16-02561-f005:**
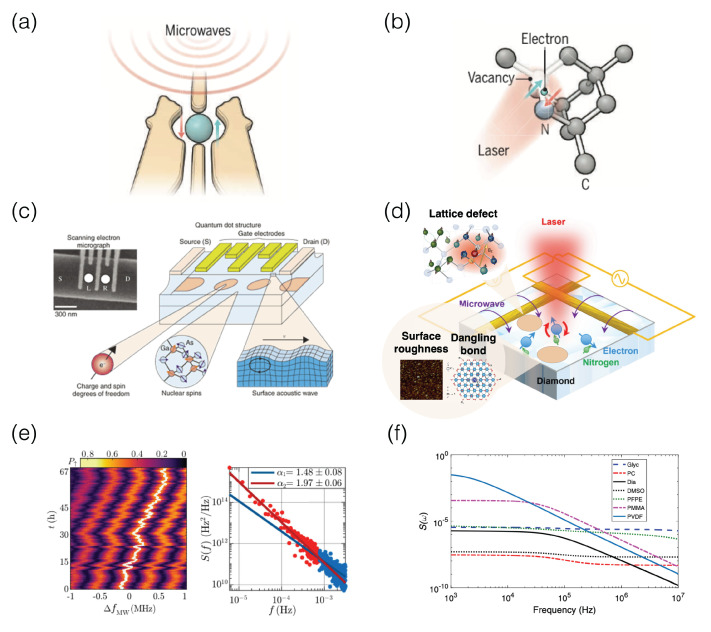
Illustration of two semiconductor spin-qubit forerunners with typical controlled schemes: (**a**) a gate-defined quantum dot (QD) with microwave and (**b**) a nitrogen-vacancy (NV) center in diamond with laser. Images are from https://www.science.org/content/article/scientists-are-close-building-quantum-computer-can-beat-conventional-one, accessed on 20 August 2022 (public domain). (**c**,**d**) Sketch of known noise sources in QD and NV center devices, respectively. Image of (**c**) is reproduced from Ref. [115]; ©2008 Nippon Telegraph and Telephone Corporation. Image of (**d**) is from https://www.eurekalert.org/multimedia/943791 (public domain) and http://qeg.mit.edu/research.php, accessed on 20 August 2022 (public domain), adapted with permission from Ref. [116]; ©2021 American Physical Society. (**e**) (Left) Time evolution of the probability to be in the spin-up state over 67 h for the microwave resonance frequency detuning parameter $\Delta f_{\mathrm{MW}}$. (Right) Power spectral density of the qubit detuning with two separate $1/f^{\alpha }$ power-law regions of exponents $\alpha_{1}$ = 1.48 (blue) and $\alpha_{2}$ = 1.97 (red). Images are adapted with permission from Ref. [117]; ©2020 Springer Nature Limited. (**f**) Comparisons of power spectral densities of NV centers in six different protective materials and their influence on noise processes, adapted with permission from Ref. [118]; ©2018 American Physical Society.

**Table 1 materials-16-02561-t001:** Comparison of the frequency-dependent noise sources and properties.

Frequency	Low	Middle	High
	(f<∼MHz)	(MHz <f< GHz)	(f>∼GHz)
Dominant noise types	1/*f* noise	Shot noise	Quantum noise
	Random telegraph noise	Thermal noise	
Power spectral density trend	$\propto \frac{1}{f^{\alpha }}$	flat	∝f
	(0.5<α<2.5)	no *f*-dependence	
Origins of noise	Not specified, natural	Discretized carrier transport	Quantum fluctuations
	Interface cleanness	Agitation by thermal energy	
	Nuclear and charge fluctuations		

**Table 2 materials-16-02561-t002:** Summary of qubit platforms involved in solid-state materials.

	Superconducting (SC)	Quantum Dot (QD)	Color Center	Surface Trapped Ions
Qubit system	Energy states in Josephson Junctions	Spin states of charge carriers	Spin or energy states of charge carriers	Energy states of atomic ions
Conventional material	Al/Al${}_{2}$O${}_{3}$	GaAs/AlGaAs, Si	NV, SiV in diamonds	Al-Cu electrodes
Noise sources	Two-level fluctuators Quasiparticle tunneling Resistive loss at the interface	Nuclear spins near QD Charge traps at the interface	Undesirable defects Dangling bonds Rough surface	Electric-field noise
Remedy	Epitaxy growth Oxygen doping Film-thickness engineering	Highly purified ${}^{28}$Si Holes instead of electrons	Improve surface treatments	Ar treatment Superconductors
New material	Nb, Ta	Si/SiGe, Si-MOS, Ge/SiGe	SiC, nanodiamond	Nb, YBCO
$T_{1}$	0.36 ± 0.01 ms ^1^ 1.20 ± 0.03 ms ^2^	2.6 s ^4^ 32 ms ^5^	>10 s ^9^ 8 h ^10^	1 ms ^13^ ∞ ^14^
$T_{2}$	0.38 ± 0.01 ms ^1^ 1.48 ± 0.13 ms ^2^	28 ms ^6^ 99 ± 4 μs ^7^	580 (210) ms ^9^ 462 μs ^11^	50 s ^14^
$F_{1}$ ${}^{†}$	0.99991(1) ^2^	0.9959 ^6^ 0.99926 ± 0.00002 ^7^	0.999952 (6) ^12^	0.999999 ^14^
$F_{2}$ ${}^{§}$	0.9948 ± 0.0004 ^3^	0.947 ± 0.008 ^8^	0.9920 (1) ^12^	0.999 (1) ^15^

^†^ Single-qubit gate fidelity; ^§^ Two-qubit gate fidelity. ^1^ The best *T*_1_ time and the *T*_2,CPMG_ with a Carr-Purcell-Meiboom-Gill (CPMG) dynamical coupling pulse sequence from a transmon qubit with a new material, tantalum [112]. ^2^ An averaged energy relaxation *T*_1_ time and the highest Ramsey coherence time *T*_2_ of a fluxonium qubit [165]. ^3^ A controlled-Z gate fidelity in a superconducting quantum circuit formed by two Xmon qubit and a transmon-type tunable coupler via a new adiabatic process [166]. ^4^ The electron spin lifetime in a Si QD at the lowest magnetic field, 1.25T [167]. ^5^ The best spin relaxation time *T*_1_ for a single-hole Ge QD [168]. ^6^ The *T*_2,CPMG_ time (the Ramsey dephasing time $T_{2}^{*}$ is 120 μs) and the Clifford single-qubit gate fidelity for a ^28^Si QD [169]. ^7^ The *T*_2,CPMG_ time ($T_{2}^{*}$ is 20 μs) and the average single-qubit gate fidelity for a ^28^Si/SiGe QD [140]. ^8^ A two-qubit Clifford gate fidelity *F*^2^ in a Si double-QD system [170]. ^9^ A longitudinal relaxation time *T*_1_ (*T*_1_∼ 6.0(4) ms at room temperature) and the maximum CPMG *T*_2,CPMG_ time at 77 K of a NV^−^ center in an isotopically pure (0.01% ^13^*C* diamond ((*T*_1_ ∼ 6.0(4) ms and *T*_1_ ∼ 3.3(4) ms at room temperature) [171]. ^10^
*T*_1_ in a NV^−^ center in diamond embedded in a superconducting 3D resonator [172]. ^11^ The *T*^2^ with XY8-4 pulse-sequence of a NV^−^ center in nanodiamond at room temperature [173]. ^12^ The average single-qubit gate fidelity *F*_1_ and the two-qubit controlled-NOT gate fidelity *F*_2_ from a NV center in bulk diamond at room temperature [174]. ^13^
*T*_1_ due to the motional heating in a trapped ^40^Ca^+^ ion at 50 μm above the Al-Cu surface trap [175]. ^14^ A memory coherence time $T_{2}^{*}$ and a single-qubit gate fidelity of a ^43^Ca^+^ in a microfabricated surface trap at room temperature [154]. ^15^ A two-qubit gate fidelity of a ^43^Ca^+^ in a microfabricated surface trap made of gold on a sapphire substrate at room temperature [155].

## Data Availability

Not applicable.

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
