# Peer review of "Material-Inherent Noise Sources in Quantum Information Architecture"

_materials, 2023, doi:10.3390/ma16072561_

Round 1

Reviewer 1 Report

I have read with care and interest this long review which has a double aim. First of all, to offer the reader an overview of the current level of theoretical understanding of the physical origin  of sources of noise that afflict the operation of any quantum device designed to exploit the peculiar behaviour  of the quantum world  to generate and sustain  quantum information protocols and quantum computation.

Secondly, to combine this analysis with a description, not from a bird's eye view, of the state of the art of relative technological progress, highlighting often very sophisticated efforts to create scalable devices that can exploit principles of quantum mechanics by working reliably, i.e. effectively counteracting intrinsic sources of quantum coherence losses. 

 I immediately claim  that reading the manuscript (perhaps renouncing to do it all in a row) is pleasant and instructive. I appreciate  that there are no  hyperbolic declarations of being close  to the achievement of commercially and scientifically gratifying and revolutionary goals placed in the foreground.
With these premises it goes without saying that my opinion is positive.
But before  writing my imaginable suggestion to the Editor, I would like to propose to the authors some enrichments of their manuscript that, although they might appear a little out of the way as indicated in the title and in the abstract, would certainly capture the attention of readers, myself included, if presented with the same care  with which the other sections were prepared.
I will limit myself to two points that are somehow connected.

I immediately say that the reading of the manuscript (perhaps renouncing to do it all in a row) is pleasant and instructive. There is a lack of temptations for hyperbolic declarations of efficiency achieved and with intellectual honesty the worries of theorists, experimenters and technologists for the achievement of commercially and scientifically gratifying and revolutionary objectives are placed in the foreground.

With these premises it goes without saying that my opinion is positive.

But before arriving at the imaginable suggestion to the editor I would like to propose to the authors some enrichments that although they might appear a little out of the way as indicated in the title and in the abstract, would certainly capture the attention of readers, myself included if they were presented. with the same level of progress with which the other sections were prepared.

I will limit myself to two points that are somehow connected.

1) The intrinsic noise of the materials used in the many platforms created, in particular those well described by the authors, rapidly degrades the quantum coherences of those few logical units with which one has to deal in practice for the time being. This might suggest a net worsening if, with the same noise level, it were possible to scale the number of logical units in the same physical scenario and technological realization. However, there are theoretical studies and experimental findings of situations, not immediately related to the quantum information area, in which noise induces collective effects in the matter that strengthen the temporal stability of the quantum coherence.
It would be very enriching to add a section that reviews situations of this type in the context of interest for the manuscript, looking for an answer to the question: can noise become an ally of quantum coherence in the presence of a large number of possibly interacting logical units?
2) The authors affirm the interest of the problem they faced in fields currently in great expansion revolving around Quantum Biology. Unfortunately at the points where they mention it, no bibliographic references are given, which disappoints the readers and makes the mention itself useless.
Now the on fashion theory of complexity applied to living matter, for example to organs such as the brain,  has greatly intrigued the very fathers of quantum mechanics, since the beginning of the last century. Today, a hundred years later, there is an awareness that the functioning of the brain of a living being, at the macroscopic level, requires mechanisms that can only be explained in quantum terms.
It is not bizarre to think that the coherences, whatever they mean, of the brain processes provide the mechanisms that allow the various brain districts to recognize each other and work in harmony and appropriate synchrony. The question I consign to the  author's reflections is this: don't you think that something in the brain resembling what we call noise could  play a role in  the macroscopic functioning of the brain, in particular to understand its ability to react   to external stimuli or vital needs,  that is making possible life ?  I am aware that this digression may appear bizarre and perhaps not well motivated in the context of a report. But I do not want to miss the opportunity to address the authors trying to convince them that my suggestion, if accepted even in a problematic form or general reflection, would make the enamel of the manuscript shinier and more brilliant 
My request is of course  optional except for the inclusion of an adequate bibliography concerning Quantum Biology or Medicine in the points where it is mentioned. If the authors do not want to do this, I would like these mentions  deleted for the reasons set out above.
In conclusion: the manuscript is well constructed, well written, instructive and useful and can be published.  However, it could become much better because it could induce other papers  ( therefore collecting  many citations) or reflections of great interest  establishing a pregnant contact with  the complexity theory that gave the Nobel Prize 2022 for physics to prof. Parisi.

Reviewer 2 Report

It is an interesting and informative overview of problems with noise in practical quantum computing from experts in the field. It is definitely worth publishing. I  am even   for publishing the manuscript as it is.

However, I cannot refrain from pointing out that as a review this manuscript could really be improved. It heaps useful and interesting practical details about recent developments. But it provides neither  good background of noise origin and description, nor wide comprehensive perspective. There are some vague general statements more appropriate in a tutorial or popular piece, and then one sees quite a high-level discussion aimed for the experimenters. Several themes are snatched and discussed.  To add, there is a piece about multilevel random telegraph noise, which seems to be out of its proper place in a review. 

To conclude: yes, it could be published.  But as the review it seems a bit lacking. 

Reviewer 4 Report

This work reviews the progress on the material issues of quantum technologies. The contents look scientifically correct and comprehensive. Therefore I recommend the work published on Materials.

Author Response

Thank you for your support. Please find our revised manuscript.